# Hydrated Zinc Borates and Their Industrial Use

**DOI:** 10.3390/molecules24132419

**Published:** 2019-06-30

**Authors:** David M. Schubert

**Affiliations:** AvidChem LLC, Lone Tree, CO 80124, USA; david.schubert@avidchem.com

**Keywords:** borate, zinc, polymer additive, fire retardant, bio-composite

## Abstract

Zinc borates are important chemical products having industrial applications as functional additives in polymers, bio-composites, paints and ceramics. Of the thirteen well documented hydrated binary zinc borates, Zn[B_3_O_4_(OH)_3_] (2ZnO∙3B_2_O_3_∙3H_2_O) is manufactured in the largest quantity and is known as an article of commerce as 2ZnO∙3B_2_O_3_∙3.5H_2_O. Other hydrated zinc borates in commercial use include 4ZnO∙B_2_O_3_∙H_2_O, 3ZnO∙3B_2_O_3_∙5H_2_O and 2ZnO∙3B_2_O_3_∙7H_2_O. The history, chemistry, and applications of these and other hydrated zinc borate phases are briefly reviewed, and outstanding problems in the field are highlighted.

## 1. Introduction

Zinc borates rank in the top ten boron-containing industrial chemicals in terms of global production and use [1]. Tens of thousands of tons of zinc borates are used annually in various applications utilizing their special properties. The family of binary zinc borates of general composition aZnO∙bB_2_O_3_∙cH_2_O, where c is >0, contains at least thirteen unique crystalline compounds; the most important these is 2ZnO∙3B_2_O_3_∙3H_2_O, or Zn[B_3_O_4_(OH)_3_], which is usually referred to in commerce as 2ZnO∙3B_2_O_3_∙3.5H_2_O, owing to an early error in characterization. Major applications of zinc borates include lending durability to bio-composite building materials and improving fire performance and electrical properties of polymers. Zinc borates also serve as corrosion inhibitors, fire retardants and preservatives in coatings, fluxes in ceramic bodies and glazes, hosts for scintillation compounds, and ingredients in agricultural micronutrients. Here the chemistry of hydrated binary zinc borates is reviewed, with emphasis on those of industrial scale commercial interest. Applications of industrial zinc borates are discussed, including their modes of action as a preservative in bio-composite building materials and as fire retardants. Outstanding problems in the field are highlighted.

Table 1 lists crystalline hydrated zinc borates that are well-documented in the chemical literature. These are arranged according their boron richness, as defined by their formal B_2_O_3_/ZnO mole ratios, or Q value. The resolved oxide formulas are used interchangeably in this review because these are widely recognized in the commercial context and can also be used to approximate compositions of zinc borates that are yet to be structurally characterized. Oxide formulas are used almost exclusively in technical literature relating to industrial applications of zinc borates as well as in regulatory documents.

Of the thirteen zinc borates listed in Table 1, five have been manufactured and sold in truckload quantities. These are described briefly in the next section. Although most crystalline zinc borates have well defined compositions and structures, a few are yet to be fully characterized including some that have been manufactured on commercial scale. While this review focuses on hydrated binary zinc borates, which have the greatest commercial activity in terms of production volumes, it can be mentioned that several crystalline anhydrous zinc borates have been structurally characterized, including Zn_4_O(BO_2_)_6_ (4ZnO∙3B_2_O_3_) [2], Zn_3_(BO_3_)_2_ (ZnO∙B_2_O_3_) [3] and ZnB_4_O_7_ (ZnO∙2B_2_O_3_) [4]. These have applications in glass making and electro-optical material. The chemistry of metal and non-metal borates in general has been reviewed [5].

## 2. Industrial Zinc Borates

### 2.1. 2ZnO∙3B_2_O_3_∙3H_2_O (Q = 1.5) or Zn[B_3_O_4_(OH)_3_]

This well-established commercial compound is manufactured today in much greater quantities than other zinc borates. Of the zinc borate phases, it is usually more practical for use in most applications owing to its relatively high dehydration onset temperature (ca. 290 °C). It is also the only zinc borate for which certain brands carry biocidal registrations, including with the U.S. EPA and Canadian PMRA. It is usually referred to in commerce as 2ZnO∙3B_2_O_3_∙3.5H_2_O. It was first described in the 1960s, being independently discovered by academic researchers in Germany and industrial researchers in the United States [12,14]. A practical manufacturing process involving a reaction of zinc oxide with excess boric acid in water was developed in the 1960s by industrial chemists and multi-ton scale commercial production commenced around 1970. This zinc borate is currently produced by several manufacturers and sold under various names, including Firebrake^®^ ZB, Borogard^®^ ZB, Composibor^®^, ZB-Shield, ZB-2335, ZB-467, and others.

Lehmann and co-workers described in 1967 the synthesis of a compound of composition 2ZnO∙3B_2_O_3_∙3H_2_O and provided powder X-ray diffraction data for it matching that of the modern commercial product [12]. Industrial researchers also discovered this compound in the 1960s, but described it as 2ZnO∙3B_2_O_3_∙3.5H_2_O, a compromise between analyses suggesting compositions ranging from 2ZnO∙3B_2_O_3_∙3.3H_2_O to 2ZnO∙3B_2_O_3_∙3.7H_2_O [14]. It was not until 2003 that a single crystal X-ray diffraction study unequivocally established its composition as 2ZnO∙3B_2_O_3_∙3H_2_O and provided the structural formula Zn[B_3_O_4_(OH)_3_] [15]. By that time, the composition 2ZnO∙3B_2_O_3_∙3.5H_2_O was firmly rooted in commercial literature and continues to persist today. This slightly inaccurate composition also gives rise to other incorrect nomenclature in commercial use, such as 4ZnO∙6B_2_O_3_∙7H_2_O. All of these compositions are the same compound best described by the formula Zn[B_3_O_4_(OH)_3_].

The initial report by Lehmann, et al. states that 2ZnO∙3B_2_O_3_∙3H_2_O is formed from a 4:1 mole ratio mixture of boric acid and zinc oxide at 165 °C in a closed tube. Such hydrothermal synthesis conditions are impractical for multi-ton scale manufacture of products in this value range. Nies, et al. disclosed in a 1970 patent a method for producing this zinc borate from zinc oxide and a stoichiometric excess of boric acid in water in the presence of product seed at temperatures as low as 75 °C, establishing the preferred manufacturing method. The required temperature to form this phase is about 70 °C. While detailed manufacturing parameters for production of this zinc borate have remained largely proprietary for the past fifty years, much has now been published on methods similar to those originally described by Nies [12,20,21,22,23,24,25]. Although the reaction of zinc oxide with boric acid, shown in Equation (1), has been the method of choice for major manufacturers for decades, a number of smaller producers have also made this compound by reaction of zinc salts, such as zinc sulfate, with borax and boric acid, as shown in Equation (2). This method has the disadvantage of producing byproduct salt that must be disposed, whereas the Nies process results only in byproduct water and allows the weak reaction liquor to be recycled to subsequent batches.(1)$$2\;\mathrm{ZnO}+B{\left(\mathrm{OH}\right)}_{3}\overset{\mathrm{exs}.\;B{\left(\mathrm{OH}\right)}_{3}}{\underset{H_{2}O,\;>70\;{}^{{}^{\circ}}C}{\to }}\mathrm{Zn}\left[B_{3}O_{4}{\left(\mathrm{OH}\right)}_{3}\right]+3\;H_{2}O$$
(2)$${\mathrm{ZnSO}}_{4}+{\mathrm{Na}}_{2}B_{4}O_{7}\cdot 10H_{2}O\overset{\mathrm{exs}.\;B{\left(\mathrm{OH}\right)}_{3}}{\underset{H_{2}O,\;>70\;{}^{{}^{\circ}}C}{\to }}\mathrm{Zn}\left[B_{3}O_{4}{\left(\mathrm{OH}\right)}_{3}\right]+{\mathrm{Na}}_{2}{\mathrm{SO}}_{4}+3\;H_{2}O$$

A commercially important property of Zn[B_3_O_4_(OH)_3_] (2ZnO∙3B_2_O_3_∙3H_2_O) is its relatively high dehydration on-set temperature of ca. 290 °C compared to other zinc borate phases. Most borates that crystallize from water possess either free water of crystallization or B-OH groups that condense out as water at relatively low temperatures. Zinc borate Zn[B_3_O_4_(OH)_3_] is among a small number of borates that crystallize from water under non-hydrothermal conditions at industrially practical rates and also exhibit high dehydration on-set temperatures. Only the zinc borate Zn_2_(BO_3_)OH (4ZnO∙B_2_O_3_∙H_2_O) has a higher dehydration on-set temperature (∼411 °C). and can be produced in a practical non-hydrothermal process.

Zinc borate Zn[B_3_O_4_(OH)_3_] crystallizes in the monoclinic space group P2_1_/c. Its structure is based on chains of linked cyclic B_3_O_4_(OH)_3_ fundamental building blocks, as shown in Figure 1 [15]. Tetrahedral Zn^2+^ cations are coordinated by borate B-O-B and B-OH oxygen atoms of three separate polyborate chains. A section of the polyborate chain structure in Zn[B_3_O_4_(OH)_3_] is shown in Figure 2. The structure of this zinc borate resembles that of the important industrial borate mineral colemanite, CaB_3_O_4_(OH)_3_∙H_2_O (2CaO∙3B_2_O_3_∙5H_2_O), in that both contain polytriborate chains [26]. However, the higher coordination demands of the Ca^2+^ ion in colemanite leads to the inclusion of water in the structure.

### 2.2. 4ZnO∙B_2_O_3_∙H_2_O (Q = 0.25) or Zn_2_(BO_3_)OH

This zinc borate is unique in having the highest dehydration on-set temperature of the hydrated zinc borates, i.e., ca. 411 °C, as illustrated by the thermogravimetric scans shown in Figure 3. This allows for its use in applications requiring high processing or service temperatures. It can be noted that neither 2ZnO∙3B_2_O_3_∙3H_2_O nor 4ZnO∙B_2_O_3_∙H_2_O contain free water, as indicated by their structural formulas Zn[B_3_O_4_(OH)_3_] and Zn_2_(BO_3_)OH, and dehydration results from condensation of B-OH groups at elevated temperatures.

Zinc borate 4ZnO∙B_2_O_3_∙H_2_O and methods to produce it on industrial scale were initially described in the patent literature and multi-ton scale commercial production under the tradename Firebrake^®^ 415 started in the early 1990s [9,10]. It is currently produced by various manufacturers. It is produced by reaction of zinc oxide with boric acid in water near the boiling point in the presence of product seed according to Equation (3). Boric acid must be added in stages in order to maintain a sufficiently high pH (above ca. pH 6) to allow the reaction to proceed.
(3)$$2\;\mathrm{ZnO}+B{\left(\mathrm{OH}\right)}_{3}\overset{\mathrm{seed}}{\underset{H_{2}O,\;100\;{}^{{}^{\circ}}C}{\to }}{\mathrm{Zn}}_{2}\left({\mathrm{BO}}_{3}\right)\mathrm{OH}+H_{2}O$$

As long as a seed is present, this compound crystallizes rapidly as a pure phase in practical slurry concentrations under non-hydrothermal conditions. It also forms in a reaction of zinc salts with borax in the presence of seed or without seed by slow hydrolysis of a dilute (<5%) slurry of ZnO∙3B_2_O_3_∙3H_2_O in boiling water [9,10,15].

This zinc borate crystallizes in the non-centrosymmetric monoclinic system space group *P*2_1_. Its structure was elucidated in 2000 by ab initio determination of the isostructural compound Zn_2_(BO_3_)(OH)_0.75_F_0.25_, in which fluoride occupies one quarter of the hydroxide positions [27]. A single crystal structure determination of pure Zn_2_(BO_3_)OH was reported in 2003 [11]. It has a framework structure built from two crystallographically independent ZnO_4_ tetrahedra that share corners with BO_3_ triangles with some ZnO_4_ corners occupied by hydroxyl groups.

### 2.3. 3ZnO∙3B_2_O_3_∙5H_2_O (Q = 1.00)

This zinc borate phase has been a commercial product for many years but is produced in much smaller quantities than ZnO∙3B_2_O_3_∙3H_2_O. As with most zinc borates, it is encountered as a microcrystalline powder that is not amenable to single crystal X-ray diffractions studies. Its crystal structure and precise chemical composition are currently unknown. It is often described in commercial literature as either ZnO∙B_2_O_3_∙2H_2_O or 2ZnO∙2B_2_O_3_∙3H_2_O and referred to as an article of commerce as ZB-112 or ZB-223. Analyses by this author suggest an actual composition close to ZnO∙B_2_O_3_∙1.67H_2_O, making 3ZnO∙3B_2_O_3_∙5H_2_O a more accurate resolved oxide formula. Although its composition is close to that of the well-defined compound ZnO∙B_2_O_3_∼1.12H_2_O, discussed below, it is a unique crystalline phase that exhibits a characteristic powder X-ray diffraction pattern and thermal profile. It has a dehydration on-set temperature of about 200 °C, allowing it to be processed in a variety of polymers in which it is used as a fire retardant synergist and smoke suppressant.

This zinc borate can be prepared by aqueous reaction of borax with a dilute zinc nitrate solution. A more practical method developed by industrial chemists is by a reaction of zinc oxide with boric acid in hot aqueous solution. Whereas formation of 2ZnO∙3B_2_O_3_∙3H_2_O requires excess boric in solution, this phase forms only when boric is present in close to the stoichiometrically required amount as given by Equation (4). When a substantial stoichiometric excess of boric acid is present the product contains 2ZnO∙3B_2_O_3_∙3H_2_O.
(4)$$3\;\mathrm{ZnO}+10\;B{\left(\mathrm{OH}\right)}_{3}\overset{H_{2}O}{\to }3\mathrm{Zn}O\cdot 5B_{2}O_{3}\cdot 5H_{2}O+10H_{2}O$$

Typically, a slight stoichiometric excess of boric acid is needed to avoid the presence of unreacted zinc oxide in the product. With careful control of stoichiometry substantially pure 3ZnO∙3B_2_O_3_∙5H_2_O can be produced as may be confirmed by XRD and titration analysis.

### 2.4. 2ZnO∙3B_2_O_3_∙7H_2_O (Q = 1.5) or Zn[B_3_O_3_(OH)_5_]∙H_2_O

This compound has been in commercial use since at least the 1930s and was the primary commercial zinc borate in the marketplace prior to the introduction of Zn[B_3_O_4_(OH)_3_] (2ZnO∙3B_2_O_3_∙3H_2_O) around 1970. Its low dehydration on-set temperature of about 120 °C is the primary disadvantage of this zinc borate in many commercial applications. This phase forms at room temperature in an aqueous reaction of borax with zinc salts or reaction boric acid with zinc oxide. An improved method to produce this zinc borate on industrial scale from zinc sulfate, zinc oxide and borax was described in a 1946 patent [28].

A single-crystal X-ray study was report confirming the structural formula of Zn[B_3_O_3_(OH)_5_]∙H_2_O [16]. It crystallizes in the orthorhombic space group P*mna* and contains isolated [B_3_O_4_(OH)_5_]^2−^ anions, as shown in Figure 4, coordinated to Zn^2+^ cations along with one water molecule.

### 2.5. 3ZnO∙5B_2_O_3_∙14H_2_O (Q = 1.67)

This zinc borate has also been sold commercially, but is rarely encountered today because its low dehydration onset temperature makes it impractical for many applications. As an article of commerce, it is sometimes referred to as 2ZnO∙3B_2_O_3_∙9H_2_O or ZB-239. It can be prepared by aqueous reaction of borax or boric acid with zinc salts. For example, a 1958 report of this compound describes crystallization within a few hours at 30 °C from an aqueous mixture of boric acid and zinc acetate in a 1:5 mole ratio [17]. When heated, the compound begins to dehydrate at 60 °C and loses 11 moles of water by 120 °C and all 14 moles of water by 300 °C.

The structure of 3ZnO∙5B_2_O_3_∙14H_2_O has not been determined and there is some uncertainly regarding its precise composition. It is nevertheless a unique crystalline phase having a characteristic powder X-ray diffraction pattern and thermal profile. The proposed structural formula of this compound is Zn_3_[B_5_O_6_(OH)_6_]_2_∙8H_2_O, shown in Figure 5. This is a zinc salt of the [B_5_O_6_(OH)_6_]^3−^ anion, which is found in other borate compounds including the important industrial mineral borate ulexite, NaCa[B_5_O_6_(OH)_6_]_2_∙5H_2_O.

## 3. Other Hydrated Zinc Borates

### 3.1. Overview of Other Hydrated Zinc Borates

The commercial zinc borate compounds discussed above are, or have been, produced on multi-ton scale and sold in truckload quantities. Production on this scale requires efficient and economical manufacturing methods. Although solvothermal syntheses are readily carried out on laboratory or small industrial scale, the equipment needed to undertake solvothermal production on multi-ton scale is generally too costly to be viable for chemical products in the value range of zinc borates. Therefore, the major commercial zinc borates are all produced under non-hydrothermal conditions using reaction pathways requiring at most a few hours to complete a multi-ton batch. The non-commercial zinc borates discussed in this section are prepared under solvothermal conditions often involving reaction times of days or weeks. Nevertheless, practical methods for manufacture of some of these zinc borates on industrial scale may eventually be developed, making these compounds of potential commercial interest. Some may also be of interest for smaller scale high value applications such electro-optical materials.

### 3.2. 16ZnO∙3B_2_O_3_∙3H_2_O (Q = 0.19) or Zn_8_(BO_3_)_3_O_2_(OH)_3_

This compound was described in 2006 [6]. It is prepared in about 20% yield by maintaining an aqueous mixture of the anhydrous zinc borate Zn_3_B_2_O_6_ and acetic acid in the presence of ethylenediamine in sealed tube at 170 °C for one week. It crystallizes in the non-centrosymmetric space group *R*32 and exhibits a framework structure constructed from two-ring [B_5_O_6_(OH)] fundamental building blocks and four-ring Zn_8_O_18_(OH)_3_ groups.

### 3.3. 12ZnO∙3B_2_O_3_∙H_2_O (Q = 0.25) or H[Zn_6_O_2_(BO_3_)_3_] or Zn_6_O(OH)(BO_3_)_3_

Two research groups described this zinc borate in 2006 and noted this as the correct formulation of a compound initially reported in 1993 as Zn_4_O(BO_3_)_2_ based on ab initio analysis of X-ray powder diffraction data [7,8,29]. The compound crystallizes in the rhombohedral space group R$\bar{3}$*c* and has a framework structure consisting of vertex-sharing ZnO_4_ tetrahedra and BO_3_ triangles similar to that originally described for anhydrous formulation. However, the redefined structure contains hydrogen that participates in nearly linear O–H···O hydrogen bonds. The presence of hydrogen was verified by solid state NMR spectroscopy [7].

This zinc borate can be prepared by maintaining an aqueous mixture of borax, zinc nitrate, and sodium hydroxide in a sealed vessel at 200 °C for four days [7]. Single crystals for X-ray diffraction work were prepared by heating a mixture of ZnO, B_2_O_3_, NaBr, and water in a 2:2:1:30 mole ratio at 280 °C for 20 days.

### 3.4. 6ZnO∙5B_2_O_3_∙3H_2_O (Q = 0.83)

This zinc borate was reported by Lehmann et al in 1967 [12]. It was initially prepared by heating a mixture of zinc oxide with boric acid in a 1:6–8 mole ratio with water in a sealed container for 16 hours at 165 °C. This compound also forms slowly when a dilute suspension of 2ZnO·3B_2_O_3_·3H_2_O in water is refluxed for a few weeks following initial formation of the 4ZnO·B_2_O_3_·H_2_O after about one week [15]. The compound has not been structurally characterized. It exhibits a powder X-ray diffraction pattern similar to that of the anhydrous zinc borate 4ZnO∙3B_2_O_3_ but has a dehydration on-set temperature below 100 °C and produces a distinctive IR spectrum consistent with the presence of water or hydroxyl groups.

### 3.5. ZnO∙B_2_O_3_∙∼1.12H_2_O (Q = 1.00) or Zn(H_2_O)[B_2_O_4_]∙∼0.12H_2_O

This zinc borate, first described in 2002, crystallizes from an aqueous mixture of zinc oxide, boric acid and guanidinium carbonate in a 1.0:3.3:1.0 mole ratio when heated in a sealed tube at 180 °C for 2 days [13]. The compound crystallizes in the rhombohedral system space group R$\bar{3}$*m*. It has an open architecture framework structure featuring large [B_12_O_24_]^12−^ polyborate rings. It begins to lose water when heated to about 120 °C.

### 3.6. 2ZnO∙3B_2_O_3_∙7.5H_2_O (Q = 1.5)

This is a distinct zinc borate phase that presents a powder X-ray diffraction pattern different from 2ZnO∙3B_2_O_3_∙7H_2_O. It is a higher hydrate of 2ZnO∙3B_2_O_3_∙7H_2_O having the proposed structural formula Zn[B_3_O_3_(OH)_5_]∙1.5H_2_O. It forms at room temperature under similar but slightly different conditions to the lower hydrate. In an example described by Lehmann et al., the initially formed amorphous precipitate obtained when zinc sulfate, borax, and boric acid are mixed in a 1:1:10 mole ratio in water at room temperature crystallizes after one or two days to provide this phase [12]. Zinc oxide can be substituted for zinc sulfate. This phase was sometimes found in commercial samples of 2ZnO∙3B_2_O_3_∙7H_2_O in the past.

### 3.7. 2ZnO∙3B_2_O_3_∙4H_2_O (Q = 1.5) or Zn(H_2_O)_4_[B_6_O_10_]

This zinc borate, described in 2017, crystallizes from a mixture of boric acid and zinc hydroxide in a 10:1:40 mole ratio in the presence of propanediamine when held at 220 °C in sealed tube for 10 days. It crystallizes in the orthorhombic system space group *Pna*2_1_ and has a open framework structure containing 10- and 11-membered ring channels formed from repeating cyclic [B_3_O_7_] units [19].

### 3.8. ZnO∙5B_2_O_3_∙4.5H_2_O (Q = 5)

This compound was reported by Lehmann et al. in 1967 [12]. It is obtained as a microcrystalline powder from a mixture of boric acid and zinc oxide in a >30:1 mole ratio without added water when heated in sealed container for 16 hours are 165 °C. The synthesis was repeated by this author and its initially reported powder X-ray diffraction pattern was confirmed. The compound has not been structurally characterized. It likely contains a condensed pentaborate framework structure with included water molecules. Upon heating, the compound begins to lose water at temperatures below 100 °C.

### 3.9. ZnO∙6B_2_O_3_∙5H_2_O (Q = 6) or ZnB_12_O_14_(OH)_10_

This zinc borate was reported in 2013 as a member of family of borate compounds, MB_12_O_14_(OH)_10_ (M = Mn, Fe, Zn) [19]. It was made using a two-step boric acid flux method. A mixture of zinc nitrate and oxalic acid in a 1:25 mole ratio was first heated in a sealed vessel at 150 °C for two days. The resulting product was washed with water, mixed with excess boric acid, and heated in a sealed vessel at 220 °C for another three days. The compound crystallizes in the monoclinic space group *P*2_1_/c. It has a framework structure consisting of 2D layers constructed from two kinds of triborate building blocks and interstitial zinc cations.

## 4. Applications

### 4.1. Overview of Applications

Zinc borates have been employed in commercial applications since at least the 1930s. By the 1940s, they were used extensively as fire retardants in building materials, paints, floor coverings, fire resistant fabrics, wire and cable jacketing, automotive parts, and mechanical rubber goods [30]. These applications continue to be important today and have expanded in scope. More recently developed major applications include uses as preservatives for bio-composites, anti-corrosives in coatings, fluxes in ceramics, and ingredients in agricultural micronutrients. These applications are discussed briefly below.

### 4.2. Polymer Additives

As polymer additives, zinc borates act as flame retardants, smoke and afterglow suppressants, and anti-drip and anti-arcing agents. They can also enhance thermal stability. The extensive literature on the uses of borates, including zinc borates, in fire retardancy has been reviewed [31]. This is a dynamic field with many new technical papers and patents relating appearing annually. Use of zinc borates as polymer additives dates back at least to the 1930s. Prior to 1970 the predominant zinc borates used in these applications were 2ZnO∙3B_2_O_3_∙7H_2_O and 3ZnO∙5B_2_O_3_∙14H_2_O. These were largely replaced by 2ZnO·3B_2_O_3_·3H_2_O after its introduction, as the latter can be used at the higher processing temperatures required for many newer polymer systems and production methods.

Well estabilished polymer additives applications of zinc borate 2ZnO·3B_2_O_3_·3H_2_O (called 2ZnO∙3B_2_O_3_∙3.5H_2_O in commerce) include extensive use in polyvinyl chloride (PVC), polyamides, polyolefins, epoxies, phenolics, ethylene vinyl acetate (EVA) and rubber products [32,33,34,35,36,37,38,39,40,41,42,43]. This zinc borate is used in wall coverings, automotive interiors, conveyor belts, wire and cable jacketing, carpet backings, electrical connectors, electronics assemblies, and many other items. It has also been used as an adhesion promoter in steel belted tires.

Zinc borates are used both halogenated and halogen-free polymer systems, often together mineral fillers. Antimony oxide is commonly used as a synergist for halogen-based fire retardants. Although the combination of halogen and antimony is effective in suppressing flaming combustion, it often results in excessive smoke generation under realistic fire conditions. Partial replace of antimony oxide with zinc borate in these systems usually results in substantial reduction in smoke while maintaining fire performance. Zinc borate can typically replace half of the antimony oxide normally needed as a halogen synergist and can sometimes completely replace it. In systems such as flexible PVC, this synergy is further improved by addition of alumina trihydrate.

Fire performance in halogen-free polymers in often achieved through use of mineral fillers, such as alumina trihydrate (ATH) or magnesium hydroxide (MDH), which may be used at loadings of 70% or more. When about 5% of the mineral filler is replaced by a zinc borate, fire performance may be substantially improved. This results from the ability of zinc borate to act a flux to convert the ATH or MDH into a hard, vitreous char that reduces the rate of heat release, suppresses afterglow, and improves other fire performance parameters. For example, replacement of 5% of ATH with zinc borate in non-crosslinked ethylene vinyl acetate (EVA) results in substantial reduction of the peak rate of heat release, delayed in time to ignition, and reduction in smoke generation [38,39,40].

Zinc borate 3ZnO·3B_2_O_3_·5H_2_O (often referred to as ZB-112 or ZB-223 in commerce) is thought to provide better smoke suppression in some polymer systems. Although manufactured in much smaller quantities than 2ZnO·3B_2_O_3_·3H_2_O, it is sometimes chosen for this reason. This effect might be related to its higher zinc content as 4ZnO·B_2_O_3_·H_2_O has been observed to also provide lower smoke generation in some systems.

Zinc borate 2ZnO·3B_2_O_3_·3H_2_O, which is used most widely in polymers, begins to lose water through condensation of B-OH groups at ca. 290 °C. This allows for use in many types of polymers processed at temperatures up to this temperature and even at 300 °C, since water loss is minimal in the 290–300 °C range. Increased use of engineering polymers requiring higher processing temperatures, as well as the need for higher production rates, has created a demand for zinc borates having greater thermal stability. This demand has been met in two ways. One method is by calcining hydrated zinc borates to produce anhydrous products. The other way is by use of zinc borates that have inherently higher dehydration onset temperatures. In the first case, 2ZnO·3B_2_O_3_·3H_2_O is calcined to form an anhydrous amorphous material of composition 2ZnO·3B_2_O_3_. This anhydrous zinc borate is now a well-established product in the fire retardant market. In the second case, the crystalline zinc borate 4ZnO·B_2_O_3_·H_2_O, which has a dehydration on-set temperature of about 411 °C, was developed as a commercial product in the early 1990s specifically to meet the need for higher processing temperatures. These zinc borates are used in polyamides, polyether ketones, polysulfones, fluoropolymers, polyesters and other systems requiring high processing temperatures. An advantage of 4ZnO·B_2_O_3_·H_2_O is its greater stability towards moisture compared to the more hygroscopic 2ZnO·3B_2_O_3_.

Zinc borates are also used to improve the electrical properties, such as comparative tracking index (CTI), of polymers used to make electrical connectors and related products. For example, zinc borate is used to partially replace antimony oxide or sodium antimonate in fiberglass-reinforced polyamides containing halogen fire retardants. This results in a substantial improvement in CTI, melt viscosity, thermal and color stability while maintaining UL-94 fire performance. Because polyamides and other engineering polymers are often processed at temperatures exceeding 300 °C, 4ZnO·B_2_O_3_·H_2_O and anhydrous zinc borate 2ZnO·3B_2_O_3_ are used in these systems to improve electrical properties and fire performance [32,42].

### 4.3. Preservative for Bio-Composites

Bio-composites, especially wood composites, are increasingly popular building materials. However, these materials may be susceptible to biodegradation. Zinc borate is used to lend durability to these products, including oriented strand board (OSB), laminated strand lumber, oriented structural straw board, particle board, engineered bamboo scrimber, waferboard, and wood-plastic composites [44,45,46,47,48,49].

OSB is the most common load bearing wood composite panel used in residential and commercial construction in North America. It is widely used as a primary structural sheathing material in construction for subfloors, floors, walls, roofs, structural I-beam components, and siding. It is also used for furniture frames and industrial crates and pallet tops. Global production of OSB now exceeds 2 × 10^7^ m^3^ per year [50]. Although OSB has excellent engineering properties, it may be susceptible to attack by decay fungi and insects when used under conditions favorable to wood destroying organisms. For this reason, a biocide is generally added to OSB during production. The most common biocide used in OSB is zinc borate of the form Zn[B_3_O_4_(OH)_3_] (described in the commercial context as ZnO∙3B_2_O_3_∙3.5H_2_O) for which certain brands carry required biocidal registrations. Zinc borate is typically added at 0.75–2% by weight as a fine powder to the dry wood chips. Wax and adhesive are added and the OSB panels are formed by application of heat and pressure. More than two decades of field test data now exists demonstrating long-term effectiveness of zinc borate for protection of OSB against biodegradation, making it the benchmark preservative in this market.

The mechanism by which zinc borate acts as a preservative for bio-composites involves controlled hydrolysis to gradually and reversibly release boric acid. Boric acid is well known to inhibit wood destroying organisms and carries biocidal registration for this use in many jurisdictions. However, its relatively low dehydration onset temperature (<100 °C) and high solubility makes it impractical to incorporate directly into most building materials. Zinc borate ZnO∙3B_2_O_3_∙3H_2_O exhibits both low solubility and a high dehydration temperature, making it suitable for use in bio-composite manufacturing processes.

Zinc borate ZnO∙3B_2_O_3_∙3H_2_O, which has the structural formula Zn[B_3_O_4_(OH)_3_], displays incongruent solubility in water with hydrolysis resulting in more soluble boric acid and less soluble zinc hydroxide, according to Equation (5). This hydrolysis is reversible and proceeds to a large extend only under high dilution conditions.

Zn[B_3_O_4_(OH)_3_] + 4 H_2_O ⇌ Zn(OH)_2_ + 3 B(OH)_3_(5)

A 5 wt% aqueous slurry of ZnO∙3B_2_O_3_∙3H_2_O remains essentially unhydrolyzed at room temperature, whereas dilution to 0.05% results in complete hydrolysis. Under conditions relevant to the use of zinc borate in building materials, movement of moisture during service results in limited hydrolysis zinc borate with release of boric acid only when required to prevent the growth of decay fungi [51].

### 4.4. Coatings

Zinc borate ZnO∙3B_2_O_3_∙3H_2_O is used as a corrosion inhibitor, in-can preservative, and tannin stain blocker in aqueous and non-aqueous coatings. It is also used as a component in some fire retardant and intumescent coatings [52].

### 4.5. Ceramics and Ceramic Glazes

Zinc borates are used as fluxes in ceramic bodies and as ingredients in ceramic glazes. Addition of zinc borate, most often ZnO∙3B_2_O_3_∙3H_2_O, at 1–10 wt% to ceramic bodies is reported to substantially decrease firing times and/or firing temperatures and to increase both green and fired strength. Zinc borates have been used for these reasons in vitreous porcelains, bricks, and other related products [53,54].

### 4.6. Agriculture

Adequate supplies of both zinc and boron are needed for the proper functioning of plants. Suboptimal soil concentrations of these two elements are among the most prevalent micronutrient deficiencies facing global agriculture. Tens of thousands of tons of refined sodium borates and beneficiated borate minerals are currently used to boost crops yields and prevent plant diseases related to boron deficiency. Zinc borates are also used as components in some agricultural micronutrient formulation.

## 5. Conclusions

Industrial applications of zinc borates continue to develop. Currently, more than 100 new patents and technical papers appear annually in this field. Despite the extensive and growing industrial applications of zinc borates, some outstanding questions remain unanswered regarding their chemistry. For example, phase relationships in the ZnO-B_2_O_3_-H_2_O system have not been fully defined. Conflicting data in the literature regarding conditions under which distinct crystalline phases form may result from the high susceptibility toward seeding exhibited by many zinc borates. In addition, structural elucidation is still needed for some well-documented phases, including those having approximate compositions 6ZnO∙5B_2_O_3_∙3H_2_O, ZnO∙5B_2_O_3_∙4.5H_2_O, and 3ZnO∙3B_2_O_3_∙5H_2_O. The last of these is often described in commercial use by the compositions ZnO∙B_2_O_3_∙2H_2_O and 2ZnO∙2B_2_O_3_∙3H_2_O, but these are likely inaccurate descriptions.

## Figures and Tables

**Figure 1 molecules-24-02419-f001:**
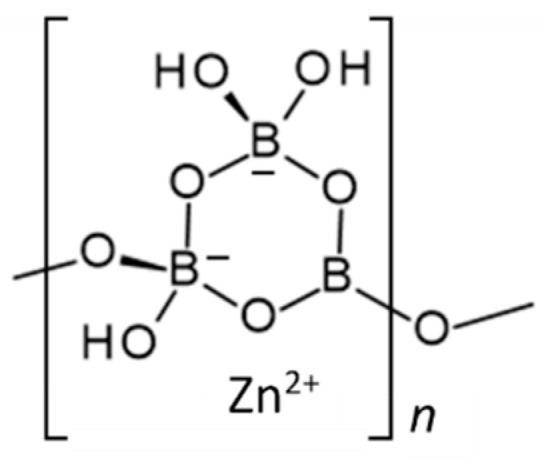
Schematic representation of the structure of Zn[B_3_O_4_(OH)_3_] (ZnO∙3B_2_O_3_∙3H_2_O) [15].

**Figure 2 molecules-24-02419-f002:**
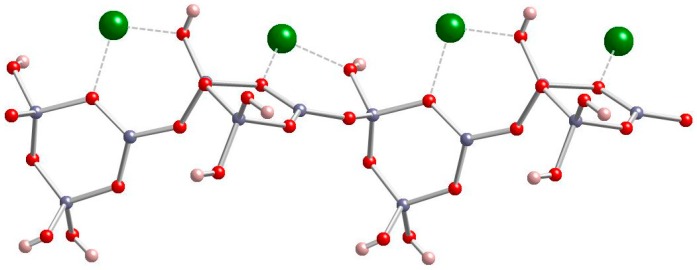
Four repeating units of the polytriborate chain in Zn[B_3_O_4_(OH)_3_]. (gray = B, red = O, green = Zn, pink = H) [15].

**Figure 3 molecules-24-02419-f003:**
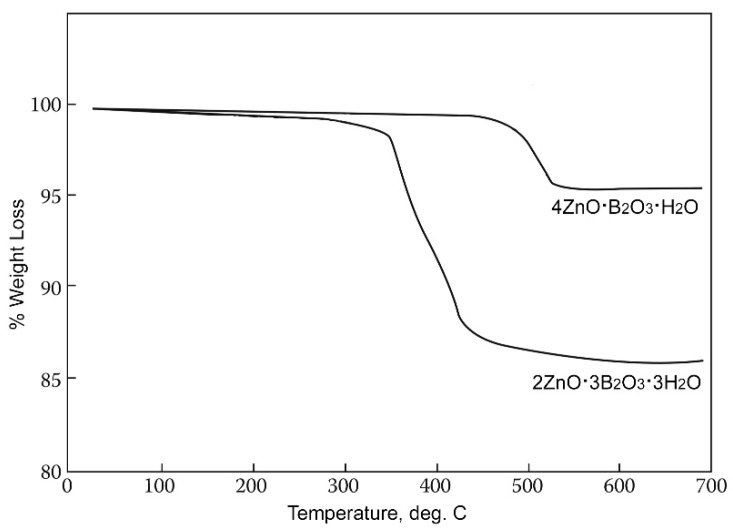
TGA scans of Zn[B_3_O_4_(OH)_3_] (ZnO∙3B_2_O_3_∙3H_2_O) and Zn_2_(BO_3_)OH (4ZnO∙B_2_O_3_∙H_2_O).

**Figure 4 molecules-24-02419-f004:**
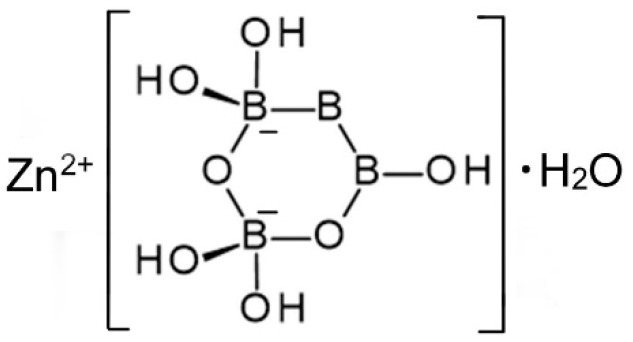
Schematic representation of the structure of Zn[B_3_O_3_(OH)_5_]∙H_2_O [13].

**Figure 5 molecules-24-02419-f005:**
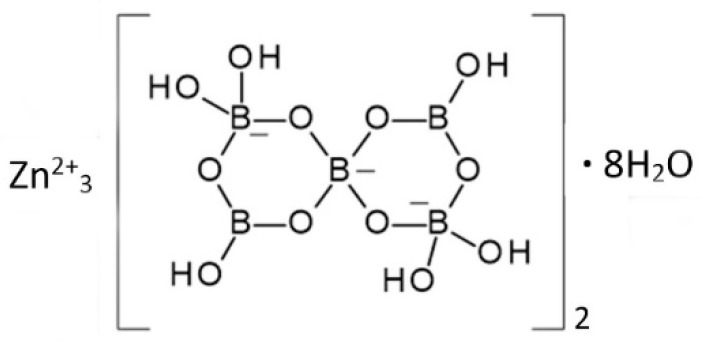
The proposed structure of 3ZnO∙5B_2_O_3_∙14H_2_O [17].

**Table 1 molecules-24-02419-t001:** Hydrated zinc borates.

Q (B_2_O_3_/ZnO)	Resolved Oxide Formula	Structural Formula	Industrial Product	Reference
0.19	16ZnO∙3B_2_O_3_∙3H_2_O	Zn_8_(BO_3_)_3_O_2_(OH)_3_		[6]
0.25	12ZnO∙3B_2_O_3_∙H_2_O	Zn_6_O(OH)(BO_3_)_3_		[7,8]
0.25	4ZnO∙B_2_O_3_∙H_2_O	Zn_2_(BO_3_)OH	yes	[9,10,11]
0.83	6ZnO∙5B_2_O_3_∙3H_2_O	unknown		[12]
1.00	ZnO∙B_2_O_3_∙∼1.12H_2_O	Zn(H_2_O)[B_2_O_4_]∼0.12H_2_O		[13]
1.00	3ZnO∙3B_2_O_3_∙5H_2_O	unknown	yes	
1.50	2ZnO∙3B_2_O_3_∙3H_2_O	Zn[B_3_O_4_(OH)_3_]	yes	[12,14,15]
1.50	2ZnO∙3B_2_O_3_∙7H_2_O	Zn[B_3_O_3_(OH)_5_]∙H_2_O	yes	[16]
1.50	2ZnO∙3B_2_O_3_∙7.5H_2_O	unknown		[12]
1.67	3ZnO∙5B_2_O_3_∙14H_2_O	Zn_3_[B_5_O_6_(OH)_6_]∙8H_2_O ^a^	yes	[17]
3.00	ZnO∙3B_2_O_3_∙4H_2_O	Zn(H_2_O)_4_[B_6_O_10_]		[18]
5.00	ZnO∙5B_2_O_3_∙4.5H_2_O	unknown		[12]
6.00	ZnO∙6B_2_O_3_∙5H_2_O	ZnB_12_O_14_(OH)_10_		[19]

^a^ proposed.

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
