# Peer review of "Hydrated Zinc Borates and Their Industrial Use"

_molecules, 2019, doi:10.3390/molecules24132419_

Round 1
Reviewer 1 Report
Dear Author,
The manuscript presents interesting results and can be published after references corrections
according to journal requirements.
4. Massa, W.; Yakubovich, O.V.; Dimitrovab, ..."
Dimitrovab should be Dimitrov A.B.
Best wishes
Author Response
Thank you for pointing out the reference error. I have corrected the authors name to Dimitrova.
In my haste to complete the manuscript I also neglected to add an important review article, Beckett, Coord. Chem Rev. 2016. This has now been added.
Reviewer 2 Report
The submitted review is very interesting and the data presented provide most valuable information to the scientific community, as the study of hydrated zink borates is a hot topic. Although, there are several issues that the authors need to address. Below are comments and suggestions:
(1) Page 4, line 122 “…neither ZnO∙3B2O3∙3H2O nor…” - based on the Fig. 3 and paragraph 2.1 correct version is 2ZnO∙3B2O3∙3H2O
(2) Page 4, line 134 “…maintain a sufficiently high pH…” Please, provide pH value with the corresponding references.
(3) It seems unnecessary to include 2ZnO∙3B2O3∙7.5H2O in the list of industrial zink borates (paragraph 2.5) based only on research of half a century ago. Its formation can results from unstable growth conditions, for example. And there are no data on its commercial use. If there are other references on this zink borate, please add them to the review.
(4) In conclusion it is emphasized that “…more than 100 new patents and technical papers appear annually…” in the field of industrial applications of zinc borates. In this regard, it would be useful to add some more recent references.
Author Response
(1) Thank you for pointing up the formula error. This has been corrected.
(2) A pH value was added
(3) Thank you for this observation. The section on 2ZnO∙3B2O3∙7.5H2O was relegated to part 3.
(4) Although many new articles and patents appear on the subject of zinc borates each year, the vast majority of these are highly specialized and pertain to arcane applications. I was unable to identify any recent ones of sufficiently general interest to be appropriate to include in this general review..
Reviewer 3 Report
This is a well-written review for zinc borates used in industry. I listed some comments which may further add values for this paper.
1) In page 3, author wrote about some inaccurate or incorrect compositions but it may be unfair or may be just historical perspectives, that some products had some composition in the past. This is just a comment and not an opinion.
2) In page 2, the explanation about “Q value” is a welcome addition.
3) Author describes each material based on the on-set temperatures. If author can add more properties, such as Table or Figures of IR or Raman data or XRD data for individual materials then this review would be great.
4) In page 5, where the fluoride comes from is uncertain.
5) Figure 6 may be missing.
6) Addition of pH condition in each material preparation would help to understand the experimental condition it produced.
7) Page 9, “electrical properties” is ambiguous and may be more specific description is necessary.
Author Response
Comment 1: The discussion about incorrect compositions is important because these incorrect composition are still widely used today in commercial practice. In fact, virtually all commercial literature and regulatory documents currently in circulation use the incorrect formulas. Therefore, it is necessary to provide this information.
Comment 2: Thank you
Comment 3: Although more detailed data could be supplied, this would be a much larger project than allowed by the 5-day turn around time for reviewer responses. Also, much of this data that I am aware of remains proprietary. Having worked for many years for a major zinc borate producer and having been intimately involved in zinc borate production process, application and characterization work, I must be very careful not to even give the perception that might be disclosing proprietary information.
Comment 4: The authors of this structure determination paper employed fluoride to obtain an isostructural compound. This nevertheless provided the structure of the pure binary zinc borate without fluoride.
Comment 5: Thank you for pointing out this error which has been corrected. There is no Figure 6.
Comment 6: Some pH detail was added on page 4. Otherwise, the same explanation as for comment 3 is offered.
Comment 7: Some specifics were added to “electrical properties”